# Griseofulvin Inhibits Root Growth by Targeting Microtubule-Associated Proteins Rather Tubulins in *Arabidopsis*

**DOI:** 10.3390/ijms24108692

**Published:** 2023-05-12

**Authors:** Yanjing Guo, Jingjing Li, Jiale Shi, Liru Mi, Jing Zhang, Su Han, Wei Liu, Dan Cheng, Sheng Qiang, Hazem M. Kalaji, Shiguo Chen

**Affiliations:** 1Weed Research Laboratory, Nanjing Agricultural University, Nanjing 210095, China; 2Institute of Technology and Life Sciences; National Research Institute, Falenty, Al. Hrabska 3, 05-090 Raszyn, Poland; 3Department of Plant Physiology, Institute of Biology, Warsaw University of Life Sciences SGGW, 159 Nowoursynowska 159, 02-776 Warsaw, Poland

**Keywords:** mycotoxin, microtubule, plant hormone, reactive oxygen species (ROS), transcriptome analysis

## Abstract

Griseofulvin was considered an effective agent for cancer therapy in past decades. Although the negative effects of griseofulvin on microtubule stability are known, the exact target and mechanism of action in plants remain unclear. Here, we used trifluralin, a well-known herbicide targeting microtubules, as a reference and revealed the differences in root tip morphology, reactive oxygen species production (ROS), microtubule dynamics, and transcriptome analysis between *Arabidopsis* treated with griseofulvin and trifluralin to elucidate the mechanism of root growth inhibition by griseofulvin. Like trifluralin, griseofulvin inhibited root growth and caused significant swelling of the root tip due to cell death induced by ROS. However, the presence of griseofulvin and trifluralin caused cell swelling in the transition zone (TZ) and meristematic zone (MZ) of root tips, respectively. Further observations revealed that griseofulvin first destroyed cortical microtubules in the cells of the TZ and early elongation zone (EZ) and then gradually affected the cells of other zones. The first target of trifluralin is the microtubules in the root MZ cells. Transcriptome analysis showed that griseofulvin mainly affected the expression of microtubule-associated protein (MAP) genes rather than tubulin genes, whereas trifluralin significantly suppressed the expression of αβ-tubulin genes. Finally, it was proposed that griseofulvin could first reduce the expression of MAP genes, meanwhile increasing the expression of auxin and ethylene-related genes to disrupt microtubule alignment in root tip TZ and early EZ cells, induce dramatic ROS production, and cause severe cell death, eventually leading to cell swelling in the corresponding zones and inhibition of root growth.

## 1. Introduction

Microtubules are non-covalent polymers of tubulins and play an important role in eukaryotic cells. Microtubules are formed by tubulin heterodimers composed of α- and β-tubulins and participate in numerous cellular processes through dynamic reorganization. In normal cells, microtubules organize the cytoplasm, nucleus, and other organelles and support the physical structure of cells. The mitotic spindle, a large dynamic microtubule assembly, plays a central role in chromosome segregation and cleavage plane alignment during cell division. In plant cells, cortical microtubules influence the shape of the cell wall by affecting the spatial organization of cellulose. Because of the necessary functions of microtubules in cells, they have been considered an important and classical target of numerous successful herbicides, such as the dinitroaniline herbicide trifluralin, as well as many drugs used as antitumor agents [1]. Dinitroaniline herbicides, a group of commercial herbicides, have typically been used to control weeds in cotton, soybean, wheat, and oilseed fields. It is well known that dinitroaniline herbicides can cause depolymerization of microtubules by binding to the unpolymerized tubulin heterodimers, resulting in serious effects on root elongation and development [2]. As early as 1971, trifluralin was discovered to inhibit root growth of maize and wheat [3]. Fernandes et al. [4] suggested that trifluralin-induced inhibition of plant growth and development was due to the disruption of microtubule polymerization and spindle apparatus formation during mitosis. Recent studies have shown that site-directed mutagenesis of α- or β-tubulin confers significant resistance to trifluralin in plants. Mutation of the Arg243 residue in α-tubulin reduced the binding efficiency between the herbicide molecule and the α-tubulin subunit and resulted in a 4- to 8-fold increase in resistance to trifluralin in *Lolium rigidum* [5].

Griseofulvin, a chlorine-containing secondary metabolite first isolated from Penicillium griseofulvin, is considered an effective antifungal agent and is widely used in dermatophyte treatment [6,7]. In fungi, griseofulvin replaces the structurally similar guanine and binds to DNA, then interferes with DNA biosynthesis and finally inhibits the mitotic process and fungal growth [8]. In addition, griseofulvin is an effective anti-cancer agent. Panda et al. [9] demonstrated that griseofulvin inhibited the proliferation of HeLa cancer cells by inhibiting mitotic spindle microtubule organization and centrosomal clustering to disrupt the cell cycle and arrest the M phase of mammalian cells. It has been suggested that griseofulvin may bind to tubulin proteins at sites common to paclitaxel or within microtubule dimers [10,11]. A previous study has shown that griseofulvin binds to both α- and β-tubulin in the mammalian brain [12]. Evidence from classical molecular dynamics simulations indicated that γ-tubulin is a possible target of griseofulvin [13]. However, some reports also indicated that griseofulvin could affect the function of microtubule-associated proteins (MAPs) rather than αβγ-tubulin subunits and subsequently inhibit microtubule polymerization in mammals, as a higher concentration of griseofulvin was required to inhibit microtubule polymerization in MAP-free microtubules [9,14]. In plants, there is limited evidence for griseofulvin. Several studies have shown that griseofulvin can inhibit root elongation in wheat, causes microtubule damage and abnormalities in spindle polarity, and exhibits similar behavior to trifluralin [15]. However, the precise target and molecular mechanism of griseofulvin in plant cells remain unclear.

Our previous experiments have shown that both griseofulvin and trifluralin inhibit plant root growth but exhibit different symptoms of damage at the root tips. In this study, we aim to investigate the following two hypotheses. First, the surgical enlargement that occurs in different zones of the root tip is mainly due to the difference in the initial zone of microtubule disruption between *Arabidopsis* seedlings treated with griseofulvin and trifluralin. Second, the physiological and molecular mechanisms of root growth inhibition induced by griseofulvin due to cell swelling and cell death are different from those of trifluralin. Therefore, the microobservation technique in combination with histochemical and cytochemical staining and genetic material expressing green fluorescent protein (GFP) fused to the microtubule-binding domain (MBD) was used to monitor root tip morphological characteristics, cell viability and morphology, production of ROS, and microtubule orientation and dynamics in *Arabidopsis* seedlings treated with griseofulvin or trifluralin. Transcriptome sequencing and gene expression analysis were also performed to reveal the molecular mechanism of root growth inhibition by griseofulvin.

## 2. Results and Discussion

### 2.1. Griseofulvin Inhibited Root Growth

The effect of griseofulvin in inhibiting fungi and mammalian cells has been widely studied, but its effect on plants is rarely mentioned. In 1954, griseofulvin was reported to cause stunted and swollen root tips in wheat [16]. Here, *Arabidopsis* Col-0 seedlings were directly cultured on 1/2 MS medium with griseofulvin, and their morphological changes were constantly observed (Figure 1A). It is clear that griseofulvin strongly inhibits seedling growth, especially root development (Figure 1B). When treated with 50 μM griseofulvin for 7 days, the root development of *Arabidopsis* almost stopped. The growth condition of seedlings under the same griseofulvin concentrations for 14 days was also observed and showed serious inhibition of root growth (see insert in Figure 1B). In addition, root length was measured to further quantify the effect of griseofulvin on root growth (Figure 1C). The result was that the root length of the seedlings decreased significantly with increasing griseofulvin concentration. When the concentration increased to 40 μM, the root length decreased to less than 1 mm. Notably, root development completely stopped in the presence of 50 μM griseofulvin.

To further confirm the effect of griseofulvin on root growth, 2-day-old *Arabidopsis* seedlings grown on the conventional medium MS were transferred to media containing griseofulvin at different concentrations and cultured for 5 days (Figure 1D). It was found that griseofulvin strongly inhibited root elongation (Figure 1E,G). Roots almost stopped growing after the seedlings were transferred to the medium containing 40 μM griseofulvin (Figure 1G). In addition, significant tissue swelling was observed in the root tips upon treatment with 40 μM griseofulvin (Figure 1F). These results are consistent with the discovery by Stokes et al. [16] that griseofulvin inhibits wheat seedling growth in both root and shoot development. This is comparable to the effect of trifluralin on plant roots.

### 2.2. Abnormal Enlargement of the Root Tips Caused by Griseofulvin Is Different from Trifluralin

In early 1971, it was discovered that trifluralin, a successful commercial herbicide, inhibited root growth of corn and wheat and caused root swelling [3]. To assess the exact effect of griseofulvin on *Arabidopsis* roots, trifluralin was used as a positive control. While *Arabidopsis* seedlings were cultured directly in 1/2 MS media containing various concentrations of griseofulvin or trifluralin at 22 °C with 100 μmol (photons) m^−2^ s^−1^ light intensity for 7 days (Figure 2A), both griseofulvin and trifluralin significantly inhibited seedling growth under the concentrations of 10 to 40 μM. As the concentration increased, the inhibition of shoot, leaf, and root development became more pronounced. It is evident that griseofulvin drastically reduced root elongation as did trifluralin (Figure 2B). Obviously, the effect of griseofulvin on root growth was a little weaker than that of trifluralin. As shown in Figure 2C, the degree of inhibition of 40 μM griseofulvin was equivalent to that of 20 μM trifluralin. Thus, griseofulvin inhibits *Arabidopsis* root growth at a higher effective dose than trifluralin.

To further observe the effect of griseofulvin and trifluralin on the tissue morphology of *Arabidopsis* root tips, two-day-old *Arabidopsis* seedlings grown on the conventional medium MS were transferred to new media containing different concentrations (0, 10, 20, and 40 μM) of griseofulvin or trifluralin (Figure 2D). After 5 days of cultivation, root tips were observed under the microscope (Figure 2E). Here, the abnormal enlargement of the root tips caused by griseofulvin and trifluralin is clearly visible. When treated with 10 μM or 20 μM griseofulvin, morphological changes in root tips are difficult to detect. However, 40 μM griseofulvin caused an abnormal enlargement of root tips that appeared to occur not in the apical meristematic zone (MZ) of trifluralin-treated roots, but mainly in its rear zones, which might be the transition zone (TZ) and elongation zone (EZ). In the following text we temporarily used TZ and EZ to state the action zones of griseofulvin and will provide further powerful evidence in the later observation of microtubules by confocal laser scanning microscope. The shape of the MZ remained normal after treatment with 40 μM griseofulvin. This result is in exact agreement with that in Figure 1F. In the case of trifluralin, 20 μM or 40 μM trifluralin obviously caused an abnormal enlargement of the root tips, especially in the apical MZ. In this situation, the root tips treated with trifluralin lost their normal apical shape. The above results show that griseofulvin and trifluralin cause abnormal enlargement of *Arabidopsis* root tips in a completely different zone.

In general, abnormal enlargement of plant tissues is associated with cell death. To verify whether griseofulvin causes cell death in different zones compared with trifluralin, cell death of the above samples was determined by trypan blue (TBD) staining. Dead cells were stained dark blue with TBD [17]. In Figure 2F, it can be seen that 10 μM griseofulvin caused slight cell death in the root TZ. As the concentration increased, the blue coloration in the root TZ became darker. At 40 μM griseofulvin, all cells already died not only in the TZ but also in the MZ of the root tips. With trifluralin, significant cell death was observed in the apical MZ after 10 μM trifluralin treatment. In the presence of 20 or 40 μM trifluralin, cell death in the apical MZ of root tips became more pronounced. These results indicate that the cell death caused by griseofulvin occurred first in the root TZ, but the initial cell death occurred in the root MZ of roots treated with trifluralin.

### 2.3. ROS Production Is an Early Event during Griseofulvin- and Trifluralin-Induced Cell Death of the Root Tips

ROS is considered a key element leading to cell death in both pathogen attack and herbicide treatment [18]. To confirm the role of ROS in the process of cell death of *Arabidopsis* root tips induced by griseofulvin or trifluralin, 3,3′-diaminobenzidine (DAB) and nitro-blue tetrazolium (NBT) staining were used to detect H_2_O_2_ and O_2_^·−^ production, respectively. In the presence of H_2_O_2_, DAB polymerizes and turns reddish-brown [19]. Here, 2-day-old *Arabidopsis* seedlings grown on the conventional medium were transferred to a new medium containing 40 μM griseofulvin or trifluralin and incubated for the indicated time before DAB or NBT staining (Figure 3A). As a result, brown deposits were visible in root tips after 40 μM griseofulvin treatment for 6 h. With increasing time up to 9 and 12 h, the oxidized DAB deposits in the root tips became darker, indicating numerous H_2_O_2_ production (Figure 3B). In plant cells, O_2_^·−^ can rapidly convert to H_2_O_2_ by superoxide dismutase (SOD) [19] Intracellular O_2_^·−^ can react with NBT to generate the dark-blue insoluble formazan [20]. The development of the NBT- O_2_^·−^ reaction products in the root tips revealed that O_2_^·−^ were detected as early as 3 h after 40 μM griseofulvin treatment, with the color continuing to darken with prolonged time to 9 h and 12 h (Figure 3B). Higher magnification of NBT-stained root tips showed that 40 μM griseofulvin induced O_2_^·−^ production as early as 1 h, which occurred mainly in the TZ and the early EZ of the root tips (Figure 3C). Data in Figure 3D showed that 40 μM trifluralin significantly induced H_2_O_2_ and O_2_^·−^ production in the apical tissues of the root tips at 3 h to 12 h after treatment. In fact, large amounts of dark-blue deposits were produced in the apical MZ of trifluralin-treated root tips as early as 1 h (Figure 3E). Obviously, the original location of ROS generation in the root tips was in good agreement with the initial occurrence of cell death (Figure 2F) and morphological damage of the root tips (Figure 2E) after griseofulvin and trifluralin treatment. Furthermore, ROS production was an early event in griseofulvin- and trifluralin-caused cell death in *Arabidopsis* roots.

To demonstrate whether ROS was involved in cell death in *Arabidopsis* roots after griseofulvin or trifluralin treatment, root tip viability was detected by fluorescein diacetate (FDA) staining. FDA fluorescence decreases as the dye leaks from dead cells, so a decrease in FDA fluorescence signal is used as an indicator of loss of cell viability [21]. As shown in Figure 4, 2-day-old *Arabidopsis* seedlings grown on conventional medium were directly transferred and incubated for 1 to 3 days in new medium containing 40 μM griseofulvin or trifluralin (Figure 4A–D), or pretreated with various ROS scavengers before incubation (Figure 4E–H). It was evident that the FDA fluorescence intensity of root tips treated with griseofulvin and trifluralin decreased significantly with increasing duration (Figure 4A–D). Remarkably, griseofulvin resulted in a visible decrease in FDA signal in the root TZ after 1 d and a strong decrease in fluorescence in the root TZ and EZ after 3 d and 5 d of treatment. However, the cells in the apical MZ of the root tips remained alive after 5 days of treatment with griseofulvin (Figure 4A). With trifluralin, the FDA signals of almost all root tip cells decreased in a time-dependent manner after 1 to 3 days of treatment (Figure 4C,D). Clearly, the mode of loss of cell viability was different in griseofulvin- and trifluralin-treated root tips. In addition, four general ROS scavengers, SOD, dimethyl thiourea (DMTU), diphenyleneiodonium (DPI), and N-acetyl-cysteine (NAC), were introduced to verify the role of ROS in cell death. SOD catalyzes O_2_^·−^ into O_2_ and H_2_O_2_ [18]. DMTU is a specific ROS scavenger [22]. DPI is an inhibitor of NADPH oxidase, and NAC is the glutathione precursor [23]. It was observed that pretreatment with SOD and DMTU significantly improved cell viability (Figure 4E,F) and eliminated morphological damage (Figure 4E) in roots treated with griseofulvin. The effects of DPI and NAC pretreatment on cell damage caused by griseofulvin were not visible (Figure 4E,F). This suggests that the cell death caused by griseofulvin was due to the production of ROS, which may have originated from O_2_. These four ROS scavengers were able to partially suppress the loss of cell viability and morphological damage caused by trifluralin in root tips (Figure 4G,H). It was demonstrated that ROS plays an important role in trifluralin-induced cell death in root tips. Therefore, it was concluded that the formation of ROS as an early event is indeed involved in griseofulvin and trifluralin-induced cell death and abnormal enlargement of root tips, ultimately leading to root growth inhibition.

### 2.4. The Initial Zone of Root Tip Cell Swelling Caused by Griseofulvin Is Different from Trifluralin

To further investigate the reason for the abnormal enlargement of root tips after treatment with griseofulvin or trifluralin, propidium iodide (PI) staining was used to detect cell morphology in different zones of *Arabidopsis* root tips. This is because PI is not able to penetrate the plasma membrane of a living cell, but only remains in the interface between the cell wall and the surrounding membrane, clearly showing the shape of each cell as red fluorescence under the confocal microscope. But a broken root cell could not prevent PI from entering the plasma membrane and became visible throughout the cell by diffuse red fluorescence [24]. Here, 5-day-old *Arabidopsis* seedlings grown on conventional medium were transferred to new medium containing 40 μM griseofulvin or trifluralin and incubated for 9 h and 24 h (Figure 5A). Then, the roots were stained with PI after light pressure and viewed under the microscope. As shown in Figure 5B, the PI stain clearly drew the shape of each cell of the mock-treated root tips. There were no abnormal morphological changes in the cells of the root tips. After 9 h of griseofulvin treatment, obvious cell disruption was observed in the root tip, although the appearance of the root tip was almost normal (Figure 5C). After 24 h, griseofulvin resulted in a remarkable enlargement in the TZ of the root tips. In addition, most of the cells in the root TZ and apical MZ had lost their boundary, as indicated by the diffuse red PI fluorescence, but the cells in the root differentiation zone (DZ) still retained their good shape (Figure 5C). This indicates that the cell destruction caused by griseofulvin occurred first in the TZ of the root tips. On the other hand, root tip cells with diffuse PI fluorescence occurred mainly in the MZ after 9 h of trifluralin treatment. After 24 h trifluralin treatment, almost all cells in the entire root tips completely ruptured and exhibited marked swelling, resulting in abnormal enlargement of root tips in the apical MZ (Figure 5D). It was certain that trifluralin damaged the root tip cells more than griseofulvin at the same treatment concentration. The damaged cells in root tips treated with trifluralin occurred first in the apical MZ, which was different from the result in roots treated with griseofulvin. This result is in good agreement with the above results on cell death and production of ROS. Thus, the difference in the abnormal enlargement of root apices between griseofulvin and trifluralin should be due to the initial ROS production in the different zones of the root apex, leading to cell swelling or destruction in different zones.

### 2.5. The Initial Zone of Microtubule Disruption in the Root Tip Cells Caused by Griseofulvin Is Different from Trifluralin

Microtubules play an important role in directional cell expansion, maintenance of cell morphology, and organization of the nucleus and organelles. Plant microtubules are composed of numerous heterodimers of α- and β-monomers that likely also contain a ring of γ-tubulin proteins [25,26]. MAPs represent all proteins involved in regulating microtubule polymerization, binding to microtubules, and stabilizing or promoting microtubule assembly [1,25]. Because griseofulvin could alter root tip cell morphology (Figure 5C), it was expected that griseofulvin could affect microtubule dynamics, as well as trifluralin. Trifluralin could depolymerize microtubules and cause the mitotic spindle not to form, leading to misalignment and chromosome segregation during cell mitosis [4]. Some studies suggest that griseofulvin targets tubulins and interferes with cell processing [12,13,27], and it could also interfere with microtubule polymerization in HeLa cells [9]. Nevertheless, its mode of action in plants is not yet clear.

To monitor the effect of griseofulvin on microtubule dynamics in root tips, specific *Arabidopsis* plants expressing a microtubule reporter were used MBD-GFP [28]. The root tip consists of three zones corresponding to the different states of the cells: meristematic zone (MZ), elongation zone (EZ), and differentiation zone (DZ). In addition, there is another transition zone (TZ) between the MZ and the EZ (Figure 6A) [29]. In the *Arabidopsis* wild-type root, the orientation of the cortical microtubules is generally loosely longitudinal in the apical MZ cells and then shifts to transverse in the early TZ cells. In the EZ, it remains predominantly transverse and then shifts to the oblique direction in the DZ cells (Figure 6A) [26].

After 1 h of treatment with 40 μM griseofulvin, microtubules on the outer cell surfaces in four different zones of the root tips of the plant MBD-GFP maintained their normal dynamics and organization (Figure 6B). After 3 h, griseofulvin caused massive dissociation of cortical microtubules in root TZ and early EZ cells and partially in fast EZ cells. Transverse microtubule alignment was barely observed, but numerous fluorescent protein particles were, indicating that the microtubules of root TZ and early EZ cells and partially of fast EZ cells were dramatically disrupted. With increasing time up to 6 and 9 h, griseofulvin caused greater dissociation of cortical microtubules in root TZ and EZ cells. It was also observed that some TZ and EZ cells began to deform (Appendix A). However, during the first 9 h of griseofulvin treatment, no visible change in cortical microtubules was observed in root MZ and DZ cells (Figure 6B and Appendix A). After 12 h of griseofulvin treatment, cortical microtubules in zones other than the root DZ were severely damaged (Figure 6B). Many cells in the apical TZ and early EZ almost lost their green fluorescence, indicating complete dissociation of cortical microtubules and cell death. Most cells in the rapid EZ exhibited randomly aligned microtubules and a twisted and swelling shape (as shown by the arrows). Some cells in the MZ also showed dissociation of cortical microtubules into small fragments that exhibited faint and diffuse green fluorescence (shown as arrows). After 24 h, root cells began to dissolve in earnest in all zones, especially in the MZ and TZ and in the early EZ cells (Appendix A). Apparently, initial microtubule dissociations in the root TZ and early EZ cells already occurred when the roots were treated with griseofulvin within 3 h, which was a later event compared with the initial ROS production (Figure 3B and Figure 6B). This is likely to be the actual and direct reason for the root swelling caused by griseofulvin in the TZ. Dissociation of cortical microtubules in the apical MZ and EZ would certainly lead to inhibition of root growth.

However, in the presence of 40 μM trifluralin, the change in cortical microtubule alignment in *Arabidopsis* root tip cells differed from that induced by griseofulvin (Appendix A). The pronounced disintegration of cortical microtubules in four zones of root tips was observed 1 h after trifluralin treatment and became more severe with increasing time up to 9 and 12 h. Some cells with dissected microtubules showed a remarkable twisted and swollen shape and even rupture of the cell wall in different zones of roots treated with trifluralin for 12 h (Appendix A). Such indiscriminate damage to all cells in whole root tips could be due to the destruction of α-tubulin, a central subunit of cell microtubules. Previous studies have shown that trifluralin destroys microtubules by binding to the α-tubulin subunit [5,30]. Undoubtedly, at the same concentration, trifluralin shows faster and more severe damage to cortical microtubules in root tips (Figure 6B and Appendix A). To clarify the detailed change process of cortical microtubules in different zones of root tips after trifluralin treatment, *Arabidopsis* MBD-GFP roots were incubated with a lower concentration (20 μM) and shorter duration (10 to 60 min). Cortical microtubules in four zone cells maintained their normal orientation after 10 min of trifluralin treatment without visible damage (Figure 6A and Figure 7). After 20 min, dissociation of cortical microtubules in apical MZ cells had occurred. After 60 min, trifluralin caused substantial disintegration of cortical microtubules not only in the apical MZ but also in the other three zones of the root tips. Significant cell breakage was also observed in the MZ, TZ, and early EZ cells of trifluralin-treated roots (Figure 7). These results indicate that trifluralin initially directs root tips to microtubules in apical MZ cells. Thus, the initial dissociation of cortical microtubules caused by griseofulvin and trifluralin occurred in the different zones, leading to cell fission and root tip swelling in the corresponding zones. All in all, our evidence suggests that the direct target of griseofulvin is not the α-tubulin protein, which is the primary target of trifluralin, because the two agents showed completely different behaviors in disrupting microtubule alignment in root tips (Figure 6 and Figure 7) [9,30]. Panda et al. [9] suggested that the microtubule polymerization inhibited by griseofulvin in mammals was due to the disruption of the function of MAPs rather than the effect on αβγ-tubulins.

### 2.6. Transcriptome Analysis of Griseofulvin- and Trifluralin-Incubated Root Tips

To reveal the molecular mechanism of action of griseofulvin, large-scale transcriptome sequencing was performed on 5-day-old *Arabidopsis* seedlings incubated with 40 μM griseofulvin or 1% DMSO (mock) for 1 h or with 20 μM trifluralin for 10 min before the first visible microtubule damage occurred. Comparative transcriptome analysis showed that 358 and 227 differentially expressed genes (DEGs) were upregulated at least twofold after griseofulvin and trifluralin, respectively (*p* < 0.05) (Figure 8A, Appendix A). Only 58 DEGs were present in both griseofulvin- and trifluralin-treated samples (Figure 8A, Appendix A). Assignment of biological process terms from Gene Ontology (GO) for these 58 genes revealed 10 overrepresented groups (*p* < 0.05). Among them, genes involved in stress, stimulus, oxidation–reduction, and cell-killing responses were significantly enriched (Figure 8B). GO term enrichment analyses (*p* < 0.05) for biological processes involving 300 genes specifically induced by griseofulvin showed that plant hormone- and ROS-related genes, especially ethylene-related genes, were significantly overrepresented (Figure 8C, Appendix A). The 169 genes specifically upregulated by trifluralin had the most enriched terms in response to secondary metabolism, oxygenated compounds, salicylic acid, and so on (Figure 8D, Appendix A). Obviously, griseofulvin mainly elicited a positive root response to ethylene, jasmonic acid, and abscisic acid. Trifluralin, on the other hand, increased the response to secondary metabolic processes of anthocyanins and flavonoids, as well as the response to salicylic acid and defense substances.

On the other hand, griseofulvin and trifluralin downregulated 563 and 333 DEGs, respectively, by at least twofold (*p* < 0.05) (Figure 8E, Appendix A). They had 53 common downregulated genes (Figure 8E), which were significantly enriched in lipid metabolism-, catabolism-, and cell-wall-related responses (Figure 8F, Appendix A). Based on the GO term enrichment (*p* < 0.05) in biological processes, 510 genes specifically repressed by griseofulvin were mainly overrepresented in the single organism process, peptide transport, cell wall modification, and cell wall component of polysaccharides groups (Figure 8G, Appendix A). However, 280 genes specifically repressed by trifluralin were significantly enriched in the hormone, especially auxin and abscisic acid, and developmental response groups (Figure 8H, Appendix A). This is in direct contrast to griseofulvin, which remarkably induced hormone-dependent genes, particularly ethylene-, jasmonic-acid-, abscisic-acid-, and auxin-dependent signaling genes (Figure 8C). Such differential responses to plant hormones may be one of the important molecular mechanisms of the microtubule disruption caused by griseofulvin and trifluralin in the different zone cells of root tips. Numerous studies have shown that plant hormones orchestrate root growth by controlling cell division, cell expansion, and cell differentiation in different zones of *Arabidopsis* roots. Briefly, auxin and gibberellin mainly promote cell division, and abscisic acid inhibits cell proliferation in the root apical MZ. In the TZ, cytokinin is essentially required for cell differentiation, which was negatively affected by auxin and gibberellin. Ethylene, auxin, jasmonic acid, and abscisic acid suppress cell elongation in the EZ [31,32].

To investigate the detailed mechanism of microtubule disruption and abnormal root enlargement caused by griseofulvin and trifluralin, a series of auxin, jasmonic acid, ethylene, microtubule, cell wall, and cellular polysaccharide-responsive genes were selected according to previous reports (Appendix A) and shown in heat maps (Figure 9A,C). Our results showed that the transcript levels of eight ethylene (ERS1, CTR1, AT1G71520, ACO1, ERF1B, ERF15, ERF114, ACO5), two jasmonic acid (bHLH041, ACX1), and nine auxin (HUP54, GH3.3, IAR3, AT2G37980, ILR1, SAUR5, SAUR51, SAUR59, SAUR76)-responsive genes were significantly upregulated by griseofulvin but not by trifluralin. In contrast, trifluralin significantly decreased the transcript levels of twelve auxin- (SAUR20, SAUR21, SAUR26, SAUR34, SAUR35, SAUR62, SAUR64, SAUR65, SAUR67, SAUR77, SHY2, ILL6), seven jasmonic-acid- (CYP94C1, CYP94B3, AOC1, AOC2, OPR3, JAR1, JAZ10) and five ethylene-responsive genes (ERF53, AT1G64380, ACS4, ERF34, ERF107) that were not significantly altered in griseofulvin-treated seedlings compared with sham treatment (Figure 9A, Appendix A). This indicates that the effect of griseofulvin and trifluralin on hormone-responsive genes was completely different, which was clearly confirmed by the qRT-PCR results. As shown in Figure 9B, the relative expression levels of five selected hormone-responsive genes, GH3.3, ILL6, CYP94C1, OPR3, and ERF53, were calculated by qRT-PCR using Actin2 as a control. Consistently, these five genes were significantly downregulated by trifluralin but upregulated by griseofulvin. Auxin is known to be important for stem cell maintenance, root meristem activity, and root zone patterning [32]. Weak auxin signaling at the root tip would result in inhibiting cell division in the MZ and promoting cell elongation in the EZ [31,32]. The number of cells in the MZ is a critical determinant of root size and growth rate because it is controlled by auxin-dependent signaling and hormonal cross-talk [33]. It is concluded that inhibition of cell division in the MZ due to low auxin-dependent gene expression may play a critical role in trifluralin-induced root growth inhibition. This may explain why trifluralin can cause the formation of ROS from root apical MZ cells, resulting in cell death and tissue swelling in the MZ during root growth inhibition (Figure 2, Figure 3D,E and Figure 5D). For griseofulvin, significant enhancement of ethylene and auxin signaling should be the primary determinant of root growth inhibition and also the actual reason for the initial triggering of ROS production in the cells of the root TZ and early EZ and the subsequent triggering of cell death and swelling in the root TZ and early EZ (Figure 2, Figure 3B and Figure 5C). Cell differentiation in the root TZ has been shown to be negatively affected by auxin [31]. Ethylene is considered a potent hormonal inhibitor of cell expansion and elongation at the root tip and stimulates auxin synthesis and transport toward the root EZ during root growth and development, where it triggers a local auxin response that leads to suppression of cell elongation [31,32].

Among the thirteen microtubule-related genes, with the exception of the β-tubulin gene TUB5, which is repressed by both griseofulvin and trifluralin, the transcript levels of four MAP genes (IPGA1, AT5G66310, AT3G51150, MAP65-1) were significantly decreased by griseofulvin but not by trifluralin. Conversely, the transcript levels of three tubulin genes (TUB8, TUA1, TUA3) and five other MAP genes (TCS1, PCAP1, PCAP2, KLCR1, KLCR2) were significantly decreased by trifluralin but not by griseofulvin (Figure 9C and Appendix A, Appendix A). The synthesis of α- and β-tubulins as basic structural subunits of microtubules is regulated by the transcript levels of the genes that encode TUA and TUB [34,35]. In conjunction with the qPCR results showing that trifluralin significantly suppresses the expression level of the TUA3 gene but not that of the MAP65-1 gene, it is clear that trifluralin inhibits plant growth by binding to αβ-tubulins. This is consistent with previous evidence of αβ-tubulin mutant and molecular interactions between trifluralin and α-tubulin [4,5]. However, tubulin-directed damage cannot be invoked as an explanation for the initial occurrence of microtubule dissociation, ROS generation, cell death, and swelling in the root apical MZ in the presence of trifluralin, because all cells from different zones in the root tips share the common microtubule structures with αβ-tubulins. In contrast, griseofulvin significantly suppressed the expression of MAP genes such as MAP65-1, IPGA1, and AT3G1150 but not tubulin genes like TUA1, TUA3, or TUB8, suggesting that griseofulvin may disrupt microtubules by targeting MAPs (Figure 9D and Appendix A). This result is consistent with previous findings that griseofulvin is most likely to target MAP rather than the structural protein tubulin [9]. Here, two MAPs encoded by AT5G66310 and AT3G51150 belong to the family of ATP-binding microtubule motor proteins. The IPGA1 protein is involved in the organization of cortical microtubules. The MAP65-1 protein, located on microtubule arrays, accumulates extensively in the root EZ and plays a critical role in root growth by promoting cell proliferation and axial expansion [36]. MAPs have been shown to decorate cortical microtubules with different patterns, regulating microtubule dynamic instability, microtubule separation, and other array assembly processes [25]. Some MAPs are important in promoting the formation of specific microtubule arrays that favor the continuation of cell division and also play a specific function in controlling auxin levels and the integration of hormone signaling [33]. Recently, it was also discovered that carrier PIN-dependent auxin transport can exert feedback control over microtubule alignment and consequently affect cell wall properties and cell shape [33]. Therefore, the disturbed balance of the complex auxin signaling network may be a reasonable explanation for the different localizations of initial ROS production, microtubule dissociation, cell death, and tissue swelling in root tips between griseofulvin and trifluralin.

In plant cells, the cell wall can also provide biophysical feedback to the orientation of microtubules in the cortex, so an undisturbed cellulose process is critical for maintaining normal microtubule orientation [37]. To further investigate the relationship between cell wall and microtubule damage in root tips treated with griseofulvin or trifluralin, thirty-one genes related to cell wall and cellular polysaccharides were also shown in Figure 9C. Twelve genes (GASA10, AT5G11420, XTH22, PGL3, ATHB-1, PHS2, CSLB4, CYCP2;1, SLK2, DGR2, ROPGAP3, BDG1) that showed no significant change in trifluralin-treated seedlings were significantly reduced in transcript levels by griseofulvin. In contrast, six genes were upregulated (AT2G20870, WAK1, ECS1, CDC20.1, AT3G01710, MADA1) and ten downregulated genes (LAC17, GRPL1, PSK2, NAC007, CRSP, IRX15-L, GAE6, CESA1, CESA3, CESA6) in trifluralin-treated seedlings showed no significant change in griseofulvin-treated samples compared with sham treatment. Interestingly, three common genes (C/VIF2, PRP4, XTH33) were downregulated by both griseofulvin and trifluralin. Further evidence for this was provided by the data of expression levels of WAK1, PSK2, GASA10, and CYCP2;1 examined in seedlings treated with griseofulvin or trifluralin by qRT-PCR compared with experimental conditions (Figure 9D). Overall, griseofulvin shows a different pattern of influence on cell-wall- and cellular-polysaccharide-related genes than trifluralin. Previous studies have suggested that cellulose-dependent cell wall integrity is an important factor in controlling the polar distribution of auxin carriers of PIN proteins [38]. Thus, this is likely a cause of the initial dissociation of microtubules in the different apical zones of roots, which is due to a differential response of auxin signaling between griseofulvin and trifluralin.

## 3. Materials and Methods

### 3.1. Plant Materials and Chemicals

*A. thaliana* wild type (Col-0) and the line expressing a microtubule (MT) reporter MBD-GFP of microtubule-associated protein 4 (MAP4) in Col-0 background (a kind gift from Dr. Zhubing Hu, Henan University) were used in this research. The details of the MBD-GFP line were previously introduced by Hamant et al. [28]. All seeds were sterilized with 70% ethanol for 3 min and washed 5 times with sterile distilled water. Seeds were germinated on half-strength (1/2) MS medium containing required substances in Petri dishes, kept at 4 °C for 3 d to break the dormancy, and then cultivated in a greenhouse with 100 μmol (photons) m^−2^ s^−1^ light intensity (16 light/8 h dark), at 22 °C and 70% humidity.

In this research, griseofulvin (C_17_H_17_ClO_6_, CAS No. 126-07-8), trifluralin (C_13_H_16_F_3_N_3_O_4_, CAS No. 1582-09-8), dimethyl sulfoxide (DMSO), trypan blue (TBD), propidium iodide (PI), fluorescein diacetate (FDA), dimethyl thiourea (DMTU), N-acetyl-cysteine (NAC), superoxide dismutase (SOD), agar (CAS No. 9002-18-0), Murashige-Skoog (MS) medium, and GAMBORG’S VITAMIN 1000X were purchased from Sigma-Aldrich (Shanghai, China). Moreover, nitro-blue tetrazolium (NBT, CAS No. 298-83-9), 3,3′-diaminobenzidine (DAB, CAS No. 91-95-2), diphenyleneiodonium (DPI), and chloral hydrate (CAS No. 302-17-0) were purchased from Beyotime (Shanghai, China). One-half-strength MS medium was prepared with 2.155% (*m*/*v*) MS powder, 3.5% (*m*/*v*) agar, and 0.1% (*v*/*v*) GAMBORG’S VITAMIN 1000X and distributed into 20 mL each Petri dish (100 mm × 25 mm), which was purchased from Fisher Scientific (Pittsburgh, PA, USA). For all experiments, griseofulvin or trifluralin was dissolved in less than 1% (*v*/*v*) DMSO. Mock treatment of 1% DMSO was used as a control.

### 3.2. Griseofulvin or Trifluralin Treatment and Observation of Seedling Phenotype

*Arabidopsis* Col-0 seeds were vertically cultivated on 1/2 MS medium containing different concentrations of griseofulvin or trifluralin or 1% DMSO (mock). After 7 or 14 days, seedling photographs of seedlings were taken with a camera (Canon G15, Tokyo, Japan), and the length of the roots was measured using a vernier caliper (ROHS HORM 2002/95/EC, Xifeng, Wuxi, China). To observe the effect of griseofulvin on root morphology, 2-day-old seedlings grown on conventional 1/2 MS medium were transferred to 1/2 MS medium containing griseofulvin or trifluralin with different concentrations and cultivated for the indicated time. The root tips were cut from the seedlings and were further observed under a microscope (Zeiss Axio Imager M2, Jena, Germany).

### 3.3. Cell Death Determination

Two-day-old Col-0 seedlings grown on conventional 1/2 MS medium were transplanted to 1/2 MS medium without or with different concentrations of griseofulvin or trifluralin and cultivated for 5 days before cell death of root were detected by TBD staining according to [17]. Briefly, seedlings were immersed in TBD solutions (0.02 g TBD, 10 g phenol, 10 glycerol, 10 mL lactic acid mixed with twice volume 96% ethanol) within a 2 mL centrifugal tube and boiled in a water bath for 1 min. Subsequently, the samples immersed in TBD solutions were kept under 25 °C, 40 rpm for 12 h. The samples were distained in chloral hydrate solutions (1 kg chloral hydrate in 400 mL H_2_O, pH 1.2) and then stored in 96% ethanol. The stained root tips were put on microscope glass slides and photographed using a stereo microscope (Olympus, BH2, Tokyo, Japan).

### 3.4. ROS Test

Hydrogen peroxide (H_2_O_2_) and superoxide anion (O_2_^·−^) were tested by in vivo staining with DAB-HCl [19] and with NBT [20], respectively. Briefly, 2-day-old seedlings grown on conventional 1/2 MS medium were transplanted on 1/2 MS medium with 1% DMSO or 40 μM griseofulvin or 40 μM trifluralin and treated for the indicated time. After treatment, the seedlings were immersed with 1 mg mL^−1^ DAB (pH 3.8) or 0.5 mg mL^−1^ NBT solution for 8 h in the dark at room temperature. Then, seedlings were decolorized in boiling ethanol (96%) for 10 min. The stained seedlings were observed and photographed under a stereo microscope (Olympus, BH2, Tokyo, Japan).

### 3.5. Detection of Root Cell Viability and Morphology

Cell viability of root tips was detected by FDA staining method according to the method of [21]. Briefly, 2-day-old seedlings grown on conventional 1/2 MS medium were transferred to 1/2 MS medium without or with 40 μM griseofulvin or 40 μM trifluralin and incubated for 1 to 5 days. The treated seedlings were washed twice with 5 mM phosphate buffer (pH 7.2) and incubated in 0.01% FDA solution for 10 min at 20 °C in darkness. The root tips were cut from the stained seedlings and put on microscope glass slides with glass coverslips (24 × 32 mm) that gently exerted pressure. The intensity of FDA fluorescence was detected with a fluorescence microscope (Zeiss Axio Imager M2, Jena, Germany) using 495 nm excitation and 500–550 nm emission.

For the experiments of ROS scavengers, 2-day-old seedlings grown on conventional 1/2 MS medium were pretreated for 1 h in 1 μM DMTU, 1 μM DPI, 1 μM NAC, or 400 U mL^−1^ SOD solution under darkness before they were transferred to 1/2 MS medium without or with 40 μM griseofulvin or 40 μM trifluralin and incubated for 5 days or 3 days. Subsequently, the treated seedlings were stained with 0.01% FDA. The stained root tips were put on microscope glass slides with glass coverslips and observed under a Zeiss Axio Imager M2 microscope.

Root cell morphology was checked using PI staining according to Hu et al. [24]. Five-day-old seedlings grown on conventional 1/2 MS medium were transferred to 1/2 MS medium without or with 40 μM griseofulvin or 40 μM trifluralin and incubated for 9 and 24 h. Seedlings were stained with 10 μg mL^−1^ PI for 1 min and mounted in distilled water. The root tips were cut from the stained seedlings and put on microscope glass slides with glass coverslips (24 × 32 mm) that gently exerted pressure. The PI fluorescence was observed by a confocal laser scanning microscope (Olympus FV3000, Tokyo, Japan) with 594 nm excitation light and 605–675 nm detection light.

### 3.6. Observation of Root Microtubules

Five-day-old seedlings of the MDB-GFP line grown on conventional 1/2 MS medium were transferred to 1/2 MS medium containing griseofulvin or trifluralin and incubated for the indicated time. Subsequently, root tips were cut from the treated seedlings using a new razor blade without wounding and put on microscope glass slides with glass coverslips (24 × 32 mm) that gently exerted pressure. An Olympus FV3000 confocal laser scanning microscope was used for MDB-GFP. The observation of microtubules was performed with the 488 nm laser line for excitation and the 500–520 nm band-pass filter.

### 3.7. RNA Extraction and qRT-PCR

Five-day-old Col-0 seedlings grown on conventional 1/2 MS medium were transferred into liquid 1/2 MS medium with 40 μM griseofulvin or 1% DMSO and incubated for 1 h or into liquid 1/2 MS medium with 20 μM trifluralin for 10 min. Total RNA from the treated seedlings was extracted using TRIzol reagent according to the manufacturer’s protocol (Invitrogen, Eugene, OR, USA). RNA was converted to cDNA using PrimeScript RT reagent Kit with gDNA Eraser (RR047A, TaKaRa, Tokyo, Japan) according to the manufacturer’s instructions. Quantitative RT–PCR (qRT-PCR) analysis was conducted using the TB Green^®^ Premix Ex Taq™ (Tli RNaseH Plus) (RR420A, TaKaRa, Tokyo, Japan) with all the primers at final concentrations of 0.2 µM in an Eppendorf real-time instrument (Mastercycler ep realplex^2^, Hamburg, Germany). The PCR program consisted of an initial denaturation step at 95 °C for 30 s, followed by 40 cycles of 95 °C for 5 s and 60 °C for 30 s, a final dissociation step at 95 °C for 15 s, 60 °C for 30 s and 95 °C for 15 s. All quantitative RT–PCR reactions were conducted with three biological replicates for each sample. The transcript level of Actin2 (At3g18780) as a reference gene was used to normalize expression data. Primer sequences for the genes used in this study are listed in Appendix A. One-way ANOVA was carried out, and means were separated by Duncan LSD at 95% using SPSS Statistics 20.0 (IBM, Armonk, NY, USA).

### 3.8. RNA-Seq Library Construction and Analysis

Five-day-old Col-0 seedlings grown on conventional 1/2 MS medium were carefully transferred to liquid 1/2 MS medium with 40 μM griseofulvin or 1% DMSO for 1 h or 20 μM trifluralin for 10 min. Three independent biological replicates were carried out. Total RNA of the treated seedling was extracted using the TRIzol reagent (Invitrogen, Eugene, OR, USA), following the manufacturer’s instruction. The purity of RNA was verified by a NanoDrop 2000 Photometer (IMPLEN, Westlake Village, CA, USA). The Qubit RNA Assay Kit in Qubit 3.0^®^ Fluorometer (Life Technologies, Carlsbad, CA, USA) was used to measure RNA concentration, and the RNA Nano 6000 Assay Kit of the Bioanalyzer 2100 system (Agilent Technologies, City of Santa Clara, CA, USA) was used to evaluate RNA integrity. RNA-seq libraries were sequenced on an Illumina platform with standard Illumina protocols at Annoroad Gene Technology Co., Ltd. (Beijing, China).

Raw RNA-Seq reads were subjected to quality check and filter to remove low-quality fragments. The cleaned reads were then mapped on the *Arabidopsis* Col-0 genome (Araport 10) (https://www.araport.org, accessed on 25 June 2021) using HISAT2 v2.1.0 (Baltimore, MD, USA, accessed on 25 June 2021) and analyzed by HTSeq (http://www.huber.embl.de/users/anders/HTSeq/doc/overview.html, accessed on 25 June 2021) to calculate raw count and FPKM (Fragments per Kilobase per Million Mapped Fragments). The normalization and differential expressed genes (DEGs) between treatments and control were analyzed using DEGseq2 v1.20.0 (http://www.bioconductor.org/packages/release/bioc/html/DESeq2.html, accessed on 25 June 2021). Genes with adjusted *p* value (*q* value) ≤ 0.05 and |Log_2_FoldChange| ≥ 1 are identified as DEGs. The Venn diagrams were generated using a public web tool Draw Venn Diagram (http://bioinformatics.psb.ugent.be/webtools/Venn/, accessed on 25 June 2021). Gene Ontology (GO) enrichment analysis was performed with a public web tool agriGO v2.0 (http://systemsbiology.cau.edu.cn/agriGOv2/, accessed on 15 April 2022) to determine the significantly enriched GO terms in the data set of biological processes in *Arabidopsis* with a significance of *p* ≤ 0.05. Heatmaps showing gene expression patterns of selected genes were generated using software HemI (Heatmap Illustrator, version 1.0, http://hemi.biocuckoo.org/faq.php, accessed on 19 March 2021) [39].

### 3.9. Accession Numbers

The information of the genes in this research can be found in the *Arabidopsis* TAIR database (https://www.arabidopsis.org, accessed on 25 June 2021) as the following accession numbers: ACTIN2 (AT3G18780), GH3.3 (AT2G23170), ILL6 (AT1G44350), CYP94C1 (AT2G27690), OPR3 (AT2G06050), ERF53 (AT2G20880), MAP65-1 (AT5G55230), TUA3 (AT5G19770), WAK1 (AT1G21250), PSK2 (AT2G22860), GASA10 (AT5G59845), and CYCP2;1 (AT3G21870).

The RNA-Seq data and their experimental description are available in the NCBI SRA database with accession PRJNA827623 (https://www.ncbi.nlm.nih.gov/bioproject/827623, accessed on 17 April 2022).

## 4. Conclusions

Our results show that griseofulvin and trifluralin have essential differences in the physiological and molecular mechanisms of root growth inhibition. Trifluralin causes microtubule-directed damage by binding to αβ-tubulins and suppresses auxin-responsive gene expression. The weakened auxin signaling leads to initial ROS production, microtubule dissociation, cell death, and swelling in the root apical MZ, inhibiting cell division in the root apical meristem, eventually leading to abnormal root tip enlargement and root growth inhibition. In the case of griseofulvin, the intense auxin and ethylene signals may be the reason for the initial occurrence of ROS burst, microtubule dissociation, cell death, and swelling in the root TZ and early EZ due to its strong negative effect on MAP genes or, in the other word, MAPs. As a result, cell elongation and root development are severely restricted, ultimately leading to growth inhibition and swelling of the root tissue. Various hormone-responsive mechanisms and distinct effects on tubulin or MAP genes might be the vital reason for the phenotype that trifluralin caused MZ swelling but griseofulvin caused TZ and EZ swelling. However, how does griseofulvin regulate auxin and ethylene-related signaling levels by affecting the function of MAP? Further studies are needed to clarify this question in the future.

## Figures and Tables

**Figure 1 ijms-24-08692-f001:**
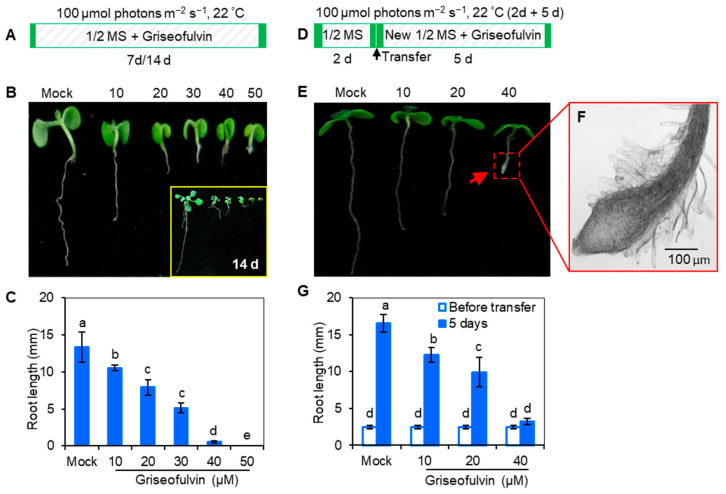
Effect of griseofulvin on the root development of *A. thaliana*. (**A**) Schematic diagram showing the strategy to monitor the effect of griseofulvin on seedling growth in *Arabidopsis*. Col-0 seeds were directly cultivated on 1/2 MS medium with 1% DMSO (mock) or griseofulvin with different concentrations (10, 20, 30, 40, and 50 μM). (**B**) Phenotypes of 7-day-old seedlings growth on the medium with griseofulvin (14-day-old seedlings shown in the inset), exhibiting a distinct concentration-dependent inhibition of root growth. (**C**) Root length of 7-day-old seedlings growth on medium with griseofulvin. (**D**) Schematic diagram showing the strategy to check the effect of griseofulvin on root growth and development in *Arabidopsis*. Two-day-old seedlings growth on the conventional 1/2 MS medium were transferred onto 1/2 MS medium with 1% DMSO (mock) or griseofulvin with different concentrations (10, 20, and 40 μM), and then continued cultivating for 5 days. (**E**) Phenotypes of the seedlings treated for 5 days with griseofulvin. (**F**) Morphological micro-observation of root tip region of 40 μM griseofulvin-treated seedlings. Scale bars: 100 μm. (**G**) Root length of the seedlings treated with griseofulvin for 5 days. Data are mean values ± SE of three independent experiments with around 30 repetitions for each treatment. Different small letters above error bars indicate significant difference at 0.05 level.

**Figure 2 ijms-24-08692-f002:**
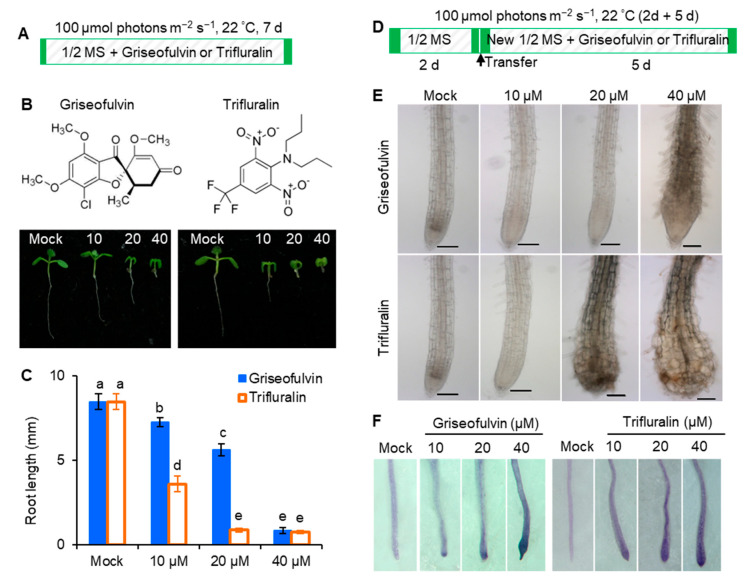
Comparison of effects of griseofulvin and trifluralin on root growth and development of *A. thaliana*. (**A**) Schematic diagram showing the strategy to monitor the effects of griseofulvin or trifluralin on root growth in *Arabidopsis*. Col-0 seeds were directly cultivated for 7 days on 1/2 MS medium with 1% DMSO (mock) or griseofulvin or trifluralin with different concentrations (10, 20, and 40 μM). (**B**) Comparison of phenotypes of 7-day-old seedlings growth on the medium with different concentrations of griseofulvin and trifluralin. (**C**) Comparison of root length of 7-day-old seedlings growth on medium with different concentrations of griseofulvin and trifluralin. (**D**) Schematic diagram showing the strategy to monitor different effects of griseofulvin and trifluralin on the morphology of root tip in *Arabidopsis*. Two-day-old seedlings growth on the conventional 1/2 MS medium were transferred onto 1/2 MS medium with 1% DMSO (mock) or different concentrations (10, 20, and 40 μM) of griseofulvin or trifluralin, and then continued cultivating for 5 days. (**E**) Morphological micro-observation of root tip region of the seedlings treated for 5 days with griseofulvin and trifluralin. Scale bars: 100 μm. (**F**) Cell death indicated by TBD staining of root tip region of the seedlings treated for 5 days with griseofulvin and trifluralin. Data are mean values ± SE of three independent experiments with around 30 repetitions for each treatment. Different small letters above error bars indicate significant difference at 0.05 level.

**Figure 3 ijms-24-08692-f003:**
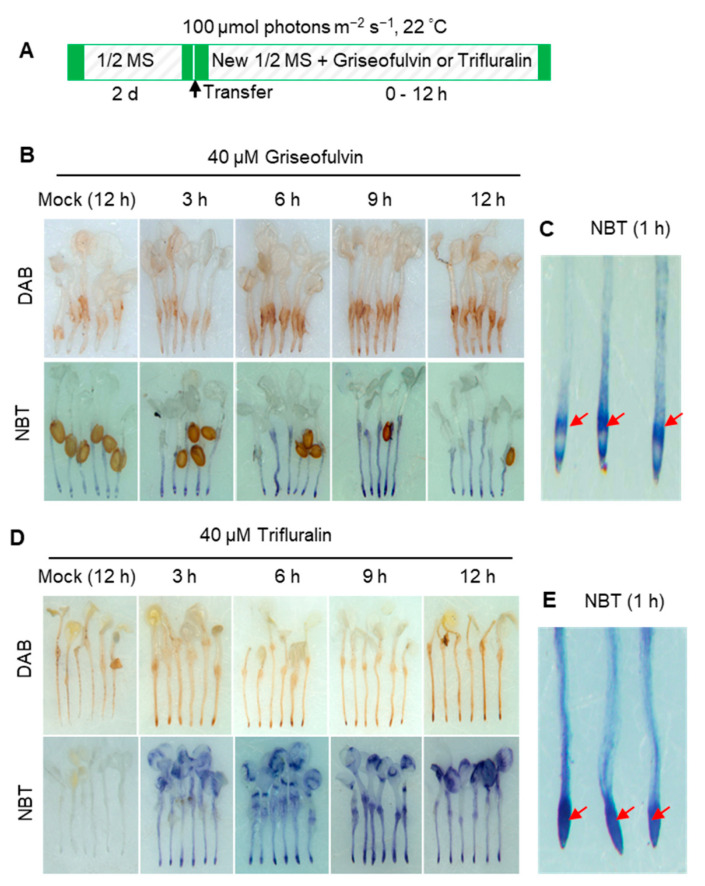
Comparison of ROS production of *Arabidopsis* roots induced by griseofulvin and trifluralin. (**A**) Schematic diagram showing the strategy to monitor ROS production of root tips induced by griseofulvin or trifluralin in *Arabidopsis*. Two-day-old seedlings growth on the conventional 1/2 MS medium were transferred onto 1/2 MS medium with 1% DMSO (mock) or different concentrations (10, 20, and 40 μM) of griseofulvin or trifluralin, and then incubated for the indicated time. (**B**–**E**) Histochemical detection of H_2_O_2_ with DAB staining and O_2_^·−^ with NBT staining in *Arabidopsis* roots treated with 40 μM griseofulvin (**B**) or 40 μM trifluralin (**D**) for 3, 6, 9, and 12 h, or treated with 1% DMSO (mock) for 12 h. NBT staining was also performed after 1 h treatment of *Arabidopsis* roots with 40 μM griseofulvin (**C**) or 40 μM trifluralin (**E**). Red arrows in (**C**,**E**) target main zone in *Arabidopsis* apical root under griseofulvin and trifluralin treatment, respectively. Results are representative of three independent experiments.

**Figure 4 ijms-24-08692-f004:**
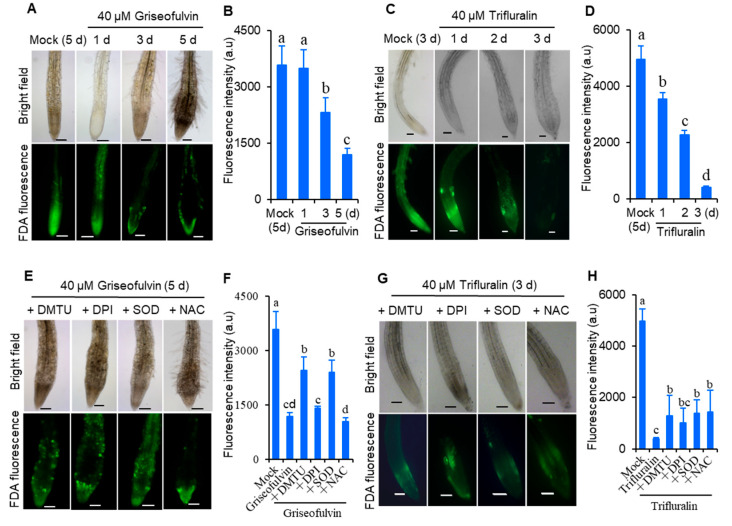
Comparison of effects of griseofulvin and trifluralin on cell viability of root tips in *Arabidopsis*. Two-day-old seedlings growth on the conventional 1/2 MS medium were transferred onto 1/2 MS medium with 1% DMSO (mock) or 40 μM griseofulvin or 40 μM trifluralin, and then incubated for the indicated time. (**A**) Images of FDA fluorescence (lower) and bright field (upper) of griseofulvin-incubated root tips. (**B**) The intensity of FDA fluorescence signals of griseofulvin-incubated root tips. (**C**) Images of FDA fluorescence (lower) and bright field (upper) of trifluralin-incubated root tips. (**D**) The intensity of FDA fluorescence signals of trifluralin-incubated root tips. (**E**–**H**) Effects of ROS scavengers on the loss of cell viability of root tips caused by griseofulvin and trifluralin. Two-day-old seedlings growth on the conventional 1/2 MS medium were pretreated for 1 h with DMTU, DPI, NAC, or SOD prior to incubation of 40 μM griseofulvin or 40 μM trifluralin. Images of FDA fluorescence (lower) and bright field (upper) of the pretreated root tips are shown after 5 days incubation of griseofulvin (**E**) or 3 days incubation of trifluralin (**G**). The intensity of FDA fluorescence signals of the pretreated root tips is calculated after incubation of griseofulvin (**F**) or trifluralin (**H**). Data are mean values ± SE of three independent measurements with at least 15 repetitions for each treatment. Different small letters above error bars indicate significant difference at 0.05 level. Scale bar: 100 μm.

**Figure 5 ijms-24-08692-f005:**
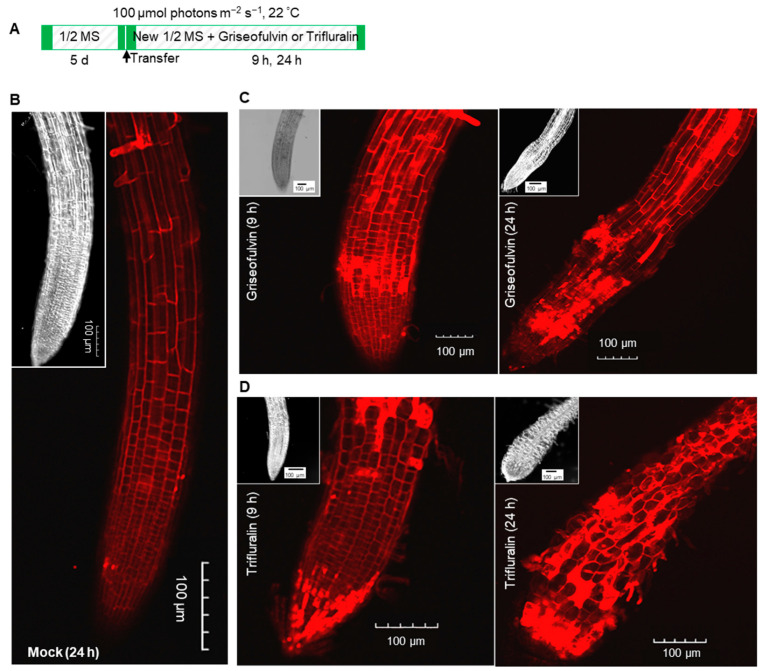
Comparison of effects of griseofulvin and trifluralin on cell morphology of root tips in *Arabidopsis*. (**A**) Schematic diagram showing the strategy to monitor the change of cell morphology of root tips induced by griseofulvin or trifluralin in *Arabidopsis*. Five-day-old seedlings growth on the conventional 1/2 MS medium were transferred onto 1/2 MS medium with 1% DMSO (mock) or 40 μM griseofulvin or 40 μM trifluralin, and then incubated for 9 and 24 h. Cell morphology of root tips was visualized using PI staining. (**B**) Cell morphology of root tips incubated for 24 h with 1% DMSO. (**C**) Cell morphology of root tips incubated for 9 and 24 h with griseofulvin. (**D**) Cell morphology of root tips incubated for 9 and 24 h with trifluralin. Images of bright field of root tips were also shown in the inset (top left corner). Results represent three independent biological replicates. Scale bar: 100 μm.

**Figure 6 ijms-24-08692-f006:**
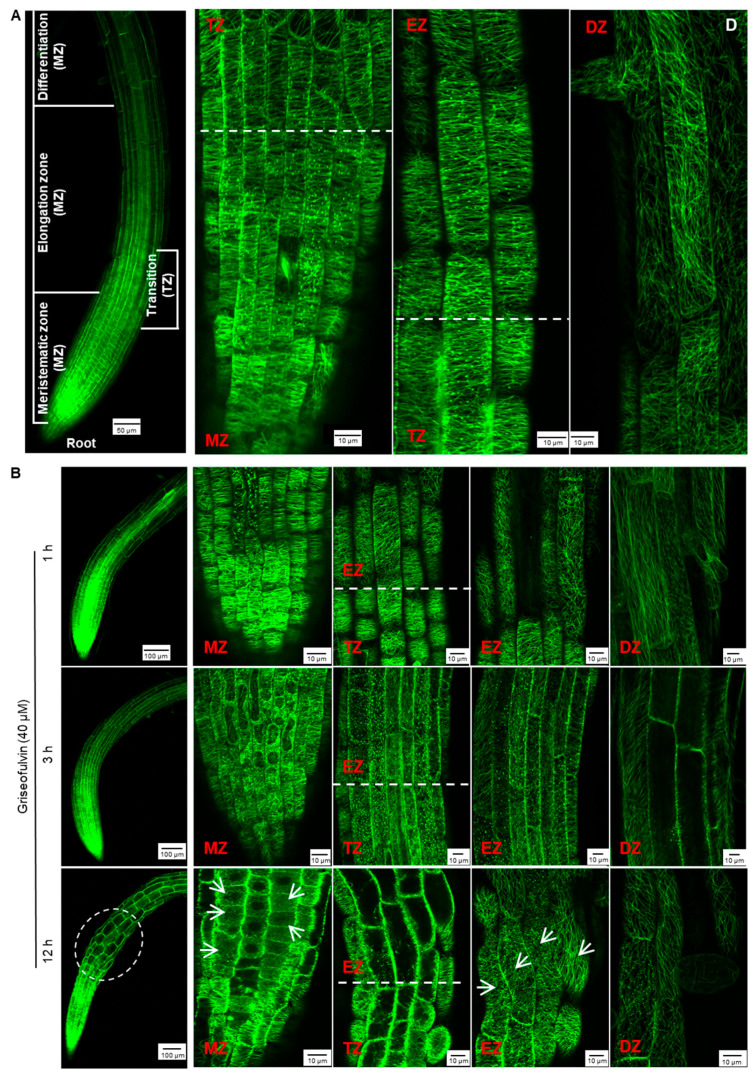
Effect of griseofulvin on microtubule dynamics of root tip cells in *Arabidopsis*. Five-day-old MBD-GFP seedlings growth on the conventional 1/2 MS medium were transferred onto 1/2 MS medium with 40 μM griseofulvin and then incubated for the indicated time. (**A**) The overall morphology (left, scale bar: 50 μm) and microtubule dynamics (right, scale bar: 10 μm) of untreated root tips. The root tips were divided into four zones, including meristematic zone (MZ), transition zone (TZ), elongation zone (EZ), and differentiation zone (DZ). Microtubules of the MZ and DZ cells showed mainly longitudinal orientation, microtubules of the TZ and EZ cells mainly showed transverse orientation. (**B**) The overall morphology (left, scale bar: 100 μm) and microtubule dynamics (right, scale bar: 10 μm) of different zones of root tips after griseofulvin incubation for 1, 3, and 12 h. White arrows point to cells with disordered microtubules. Results represent three independent biological replicates.

**Figure 7 ijms-24-08692-f007:**
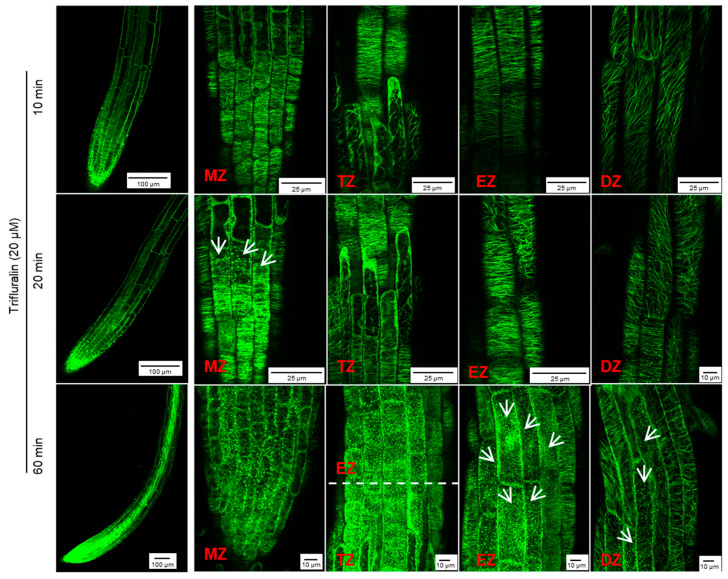
Effect of trifluralin on microtubule dynamics of root tip cells in *Arabidopsis*. Five-day-old MBD-GFP seedlings growth on the conventional 1/2 MS medium were transferred onto 1/2 MS medium with 20 μM trifluralin and then incubated for 10, 20, and 60 min. The overall morphology (**left**, scale bar: 100 μm) and microtubule dynamics (**right**, scale bar: 25 μm or 10 μm) of different zones of root tips after griseofulvin incubation were shown. White arrows point to cells with disordered microtubules. Results represent three independent biological replicates.

**Figure 8 ijms-24-08692-f008:**
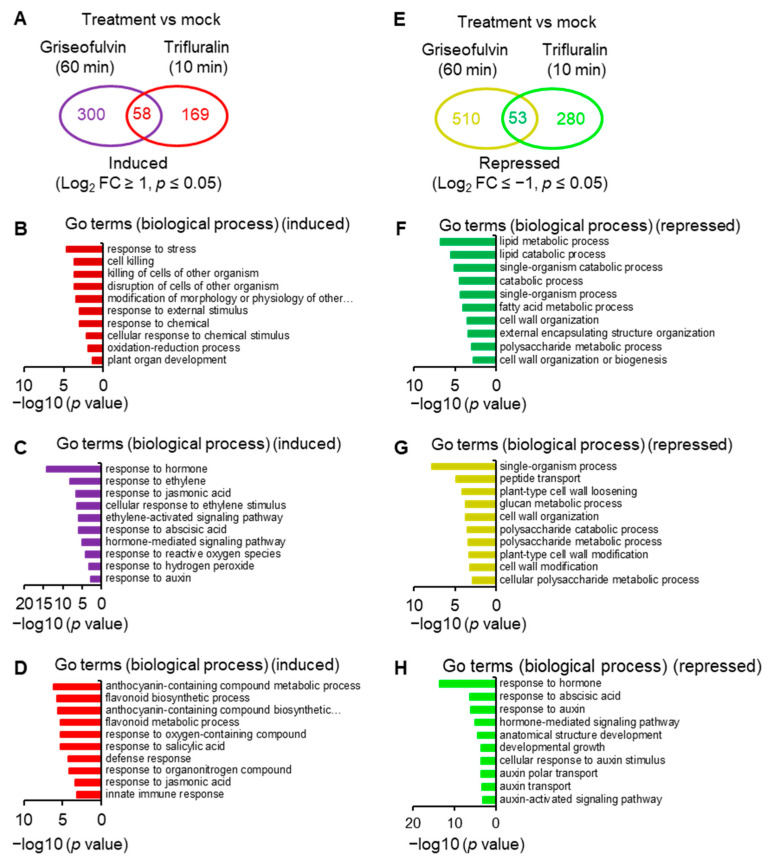
Comparative analyses of transcriptome of *Arabidopsis* seedlings incubated with griseofulvin and trifluralin. Five-day-old Col-0 seedlings growth on the conventional 1/2 MS medium were transferred onto liquid 1/2 MS medium with 1% DMSO, 40 μM griseofulvin, or 20 μM trifluralin, and then incubated for the indicated time. (**A**) Venn diagrams showing the numbers of genes that were upregulated higher than twofold in Col-0 seedlings after griseofulvin or trifluralin incubation as compared to mock treatment (1% DMSO). (**B**) Gene Ontology (GO) enrichment analysis of 58 common upregulated genes. (**C**) GO enrichment analysis of 300 genes that were only induced by griseofulvin. (**D**) GO enrichment analysis of 169 genes that were only induced by trifluralin. (**E**) Venn diagrams showing the numbers of genes that were downregulated lower than twofold in Col-0 seedlings after griseofulvin or trifluralin incubation as compared to mock treatment (1% DMSO). (**F**) GO enrichment analysis of 53 common downregulated genes. (**G**) GO enrichment analysis of 510 genes that were only repressed by griseofulvin. (**H**) GO enrichment analysis of 280 genes that were only repressed by trifluralin.

**Figure 9 ijms-24-08692-f009:**
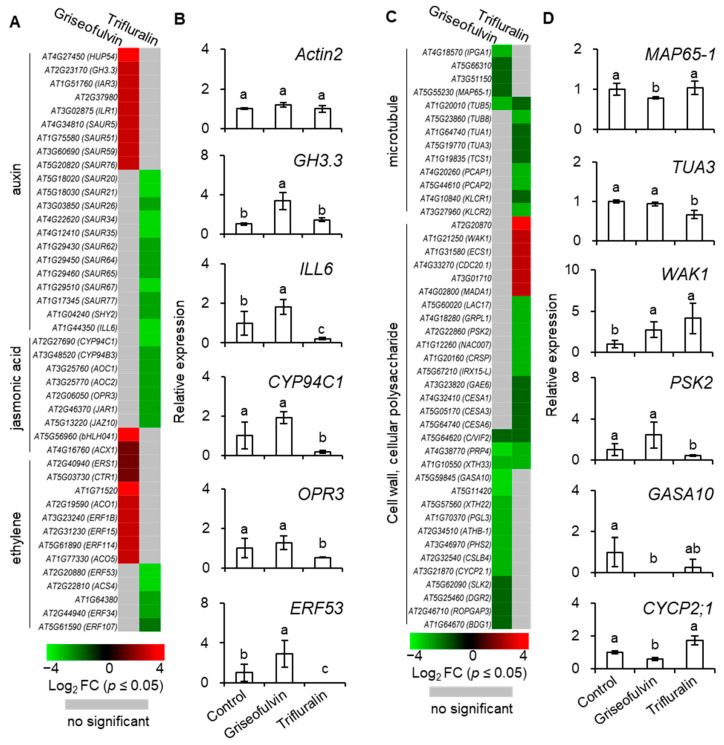
Heat map showing the expression levels of auxin, jasmonic acid, and ethylene-related response genes (**A**), microtubule and cell wall, cellular polysaccharide changes (**C**) in Col-0 seedlings inoculated by 40 μM griseofulvin for 60 min or 20 μM trifluralin for 10 min compared with mock treatment (1% DMSO). The colors of the heat map represent the Log_2_FC ranging from green (−4) through black (0) to red (4). The genes without significant difference (*p* > 0.05) were shown in gray color. The expression levels of selected plant hormone-responsive genes, including GH3.3, ILL6, CYP94C1, OPR3, and ERF53 (**B**), microtubule- and cell-wall-related genes, including MAP65-1, TUA3, WAK1, PSK2, GASA10, and CYCP2;1 (**D**), in Col-0 seedlings inoculated with griseofulvin or trifluralin were measured by quantitative PCR (qRT-PCR). Gene expression levels were normalized to ACTIN2. Data are mean ± SE of three independent biological replicates. The different small letters above error bars indicate significant difference at 0.05 level.

## Data Availability

The data presented in this study are available in the article and the Appendix A here.

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
