# Peer review of "Griseofulvin Inhibits Root Growth by Targeting Microtubule-Associated Proteins Rather Tubulins in *Arabidopsis"

_ijms, 2023, doi:10.3390/ijms24108692_

Round 1

Reviewer 1 Report

Finding new inhibitors and increasing the chemical toolbox to study cellular features is so important for cell biologists. This manuscript did a very good job in deciphering the mechanism of Griseofulvin and Trifluralin using Arabidopsis root. It will be one of fundamental study for chemical cell biology as soon as it is published and becomes available.  

Author Response

Dear Reviewer,

Our manuscript “Griseofulvin inhibits root growth by targeting microtubule-associated proteins rather tubulins in Arabidopsis” by Y. Guo et al. has been changed according to reviewers’ comments. All revised parts were marked in red color in the new version.

We would like to thank you for your kind comments. We hope that the revised version of our manuscript is able for publication in “IJMS”.

With best regards.

Reviewer 2 Report

Dear Authors,

The article "Griseofulvin inhibits root growth by targeting microtubule- associated proteins rather tubulins in Arabidopsis" is interesting research. Experimental design is not clearly provided Viz., No. of replications, reproducibility of the experiment with another set of biological replicate, etc.

Please include those data so the conclusions made from this research should be supported by statistical analysis.

Author Response

Dear Reviewer,

Our manuscript “Griseofulvin inhibits root growth by targeting microtubule-associated proteins rather tubulins in Arabidopsis” by Y. Guo et al. has been changed according to your and other reviewers’ comments. We have added statements about biological replicates in Figure legends as well as description about statistical analysis in materials and methods part. All revised parts were marked in red color in the new version. 

Thank you for your kind comments. We hope that the revised version of our manuscript is able for publication in “IJMS”.

With best regards.

Reviewer 3 Report

This study attempts to understand the mechanisms by which griseofulvin causes abnormal root growth by comparing its effects with another herbicide, trifluralin, using microscopy and transcriptomic analyses. The authors present data from a multitude of experiments but there are not strong connections between the accumulated data and data interpretation to support their conclusions.

Major changes:

11.       The authors need to add controls without DMSO to Figure 1. 1% DMSO is a fairly high concentration (most studies that I am aware of try to use 0.1% DMSO when growing seedlings). I expect 1% DMSO has negative effects on root growth and morphology. If the authors add this information in the first figure and discuss the differences in seedling phenotypes this will minimize issues later on such as what effects does 1% DMSO have on gene expression.

22.       There is a problem with Figure 2D and 2E. In order to make the claim that griseofulvin induces O2- production mainly in the TZ and EZ and that trifluralin affects apical MZ the authors need to add phase-contrast images potentially stained with PI to show the different zones of the root using an appropriate scale. However, examining the results closely to not reveal significant differences between the two different treatments as claimed. Also red arrows in Fig. 2 should be described in Figure legend.

33.       The issue described above is also a problem for Figures 3, 4 and 5. At this magnification it is difficult to claim localization in different zones of the root without seeing cell sizes or possibly using careful measurements. In Figure 6A, which is at a higher magnification you indicate where the different root zones are located but you do not indicate how this was determined? If you can provide evidence that the different root zones are properly identified for the Figure 6, then the locales of the microtubule changes will be well documented but there is still the problem of localization for the other phenotype changes presented in Figures 2-5.

44.       The letters of significance used in the bar graphs showing the q-RT-PCR data in Figure 9 needs to be corrected, control should always be designated with an A, and then different letters indicate statistically significant differences in expression. Also, a brief description of the statistical analyses should be included in the figure legend, student t-test? In the Methods section, a more thorough description of the q-RT-PCR experiment should be added. Also, for each of the 3 biological replicates the authors should have done three technical replicates, was this done but not reported?

55.       The authors need to provide an explanation for why they chose different treatment time points for the trifluralin and griseofulvin treatments for the q-RT-PCR and RNA-seq experiments i.e. why 10 min and 60 min trifluralin. This is an issue because when comparing their effects on transcript levels, we don’t actually know what the effects of one of the herbicides might have had at the other timepoint.

66.       Often the authors overinterpret their data. For example, pg 15, lines 520-523, “In conjunction with the qPCR results showing that trifluralin significantly suppresses the expression level of the TUA3 gene but not that of the MAP65-1 gene, it is clear that trifluralin inhibits plant growth by binding to αβ-tubulins.” should be rewritten “In conjunction with the qPCR results showing that trifluralin statistically significantly suppresses the expression level of the TUA3 gene but not that of the MAP65-1 gene, it is clear that one way in which trifluralin inhibits plant growth by binding to αβ-tubulins as well as slightly inhibiting tubulin expression.” Further, the authors should consider that the q-RT-PCR results show that there is only approximately a 30% decrease in TUA3 expression and approximately 20% decrease in MAP65-1 expression compared to both the control and the trifluralin-induced change in transcript levels, and these are relatively small effects on transcript levels.

77.       Thus, in the Results section, text should be added indicating that like griseofulvin, trifluralin treatment also downregulates the level of MAP gene transcripts (a different seven MAP genes than the ones that are down-regulated by trifluralin). Further, the authors should include more q-RT-PCR results of tubulin genes and MAP genes (not just one of each) in order for them to provide evidence in support of their point that differential expression of these two types of genes induced by the two different types of treatments is causative for the differences in their root phenotypes.

88.       The Discussion and Conclusions sections need careful editing. For example, the conclusions made in the Conclusion section need to be re-visited and should match better with the summary of this study provided in the Abstract section.  

Minor changes:

11.       There are certain sections of the manuscript that require careful editing by a native English speaker.

22.       In the Abstract, there is a statement “Transcriptome analysis showed that griseofulvin mainly damaged microtubule-associated protein (MAP) rather than tubulin genes” which needs correcting to “Transcriptome analysis showed that griseofulvin mainly affected expression of microtubule-associated protein (MAP) genes rather than tubulin genes”.

33.       Figure legend for 2F, the authors need to indicate that this is TBD staining.

44.       The authors need to add/brighten the scale bars in Figure 4.

55.       There are several places in the manuscript where the authors introduce an abbreviation without first giving the full name, i.e. pg 4 line 173 “TBD staining”, pg 5 line 204 “DAB and NBT staining”, and line 211 “by SOD”. Please correct this issue by indicating full length names at first mention in the manuscript followed by abbreviations. There are numerous other examples of this problem that needs correcting in the manuscript, i.e. FDA, DTMU, DPI, NAC, etc

66.       Pg 6, Line 212, change “were tested” to “was detected”

77.       Pg 6, line 215 change “The amplified version” to “Higher magnification”

88.       Figure 2 legend, does not indicate that 2F is TBD staining of roots, and does not indicate what the red arrows are doing, please add this information.

99.       Figure 5 labels need correcting, u should be changed to µ.

110.   Figure 6 scale bars should be brightened.

Round 2

Reviewer 2 Report

Dear Authors,

All the suggestions have been included and now its the Editor team's responsibility.